# Multivariate Genetic Structure of Externalizing Behavior and Structural Brain Development in a Longitudinal Adolescent Twin Sample

**DOI:** 10.3390/ijms23063176

**Published:** 2022-03-15

**Authors:** Jalmar Teeuw, Marieke Klein, Nina Roth Mota, Rachel M. Brouwer, Dennis van ‘t Ent, Zyneb Al-Hassaan, Barbara Franke, Dorret I. Boomsma, Hilleke E. Hulshoff Pol

**Affiliations:** 1Department of Psychiatry, Brain Center Rudolf Magnus, University Medical Center Utrecht, 3584 CX Utrecht, The Netherlands; r.m2.brouwer@vu.nl (R.M.B.); zyneb.al-hassaan@student.uva.nl (Z.A.-H.); h.e.hulshoff@umcutrecht.nl (H.E.H.P.); 2Department of Psychiatry, University of California San Diego, La Jolla, CA 92093, USA; mklein@health.ucsd.edu; 3Department of Human Genetics, Radboud University Medical Center, 6525 GA Nijmegen, The Netherlands; nina.rothmota@radboudumc.nl (N.R.M.); barbara.franke@radboudumc.nl (B.F.); 4Donders Institute for Brain, Cognition and Behaviour, Radboud University, 6525 XZ Nijmegen, The Netherlands; 5Department of Complex Trait Genetics, Center for Neurogenomics and Cognitive Research, Amsterdam Neuroscience, Vrije Universiteit Amsterdam, 1081 HV Amsterdam, The Netherlands; 6Department of Biological Psychology, Vrije Universiteit Amsterdam, 1081 HV Amsterdam, The Netherlands; d.vant.ent@vu.nl (D.v.‘t.E.); di.boomsma@vu.nl (D.I.B.); 7Department of Psychiatry, Radboud University Medical Center, 6525 GA Nijmegen, The Netherlands; 8Amsterdam Public Health (APH) Research Institute, 1081 BT Amsterdam, The Netherlands; 9Department of Psychology, Utrecht University, 3584 CS Utrecht, The Netherlands

**Keywords:** externalizing behavior, adolescence, gray matter volume, white matter integrity, heritability, genetic correlation, longitudinal, magnetic resonance imaging

## Abstract

Externalizing behavior in its more extreme form is often considered a problem to the individual, their families, teachers, and society as a whole. Several brain structures have been linked to externalizing behavior and such associations may arise if the (co)development of externalizing behavior and brain structures share the same genetic and/or environmental factor(s). We assessed externalizing behavior with the Child Behavior Checklist and Youth Self Report, and the brain volumes and white matter integrity (fractional anisotropy [FA] and mean diffusivity [MD]) with magnetic resonance imaging in the BrainSCALE cohort, which consisted of twins and their older siblings from 112 families measured longitudinally at ages 10, 13, and 18 years for the twins. Genetic covariance modeling based on the classical twin design, extended to also include siblings of twins, showed that genes influence externalizing behavior and changes therein (*h*^2^ up to 88%). More pronounced externalizing behavior was associated with higher FA (observed correlation *r*_ph_ up to +0.20) and lower MD (*r*_ph_ up to −0.20), with sizeable genetic correlations (FA *r*_a_ up to +0.42; MD *r*_a_ up to −0.33). The cortical gray matter (CGM; *r*_ph_ up to −0.20) and cerebral white matter (CWM; *r*_ph_ up to +0.20) volume were phenotypically but not genetically associated with externalizing behavior. These results suggest a potential mediating role for global brain structures in the display of externalizing behavior during adolescence that are both partially explained by the influence of the same genetic factor.

## 1. Introduction

Externalizing behavior is characterized by a multitude of antisocial and disruptive behaviors in the form of acting out, aggression, hostility, delinquency, or criminal acts that can be costly and potentially harmful to the individual, their family and friends, and society [1]. Externalizing behavior in children and adolescents can be assessed by the broad externalizing scale from the parent-reported Child Behavior Checklist (CBCL) and the self-reported Youth Self Report (YSR) [2]. The age of onset of externalizing behavior varies between individuals, but occurs most often during childhood and adolescence, and the trajectory of it may have different profiles of disruptive behavior throughout development [3,4,5]. Externalizing behavior occurs in both sexes, although boys are at an increased risk of disruptive behavior [6,7]. Around 5% to 10% of Dutch children and adolescents eventually receive a clinical diagnosis of oppositional defiant disorder (ODD) and/or conduct disorder (CD) [8,9,10]. Externalizing behavior can desist on its own or after successful intervention [11]. However, externalizing behavior that is allowed to escalate or continues to persist after adolescence may lead to a diagnosis of antisocial personality disorder (ASPD) or psychopathy during adulthood, especially in combination with callous-unemotional traits [12,13,14]. These psychopathic traits are elevated in early offenders and incarcerated youth [15]. As such, understanding the etiology and development of externalizing behavior during childhood and adolescence is important for developing methods for early diagnosis and treatment to prevent the escalation of antisocial behavior.

Twin and family studies have revealed a heritable factor for externalizing behavior [16,17,18,19,20,21,22,23,24,25] that is largely determined by the same genetic factor throughout child and adolescent development [26,27,28]. A recent longitudinal twin study has shown that the rate of change in the development of externalizing behavior is also in part heritable [29], but this has not been consistently reported [30]. However, it remains unclear through which biological pathways the genetic influences on externalizing behavior might act and if changes in externalizing behavior is mediated through other developmental changes occurring during adolescence, such as structural brain development [31].

The human brain continues to develop considerably during adolescence [32,33]. Although total brain volume has nearly reached its adult size by late childhood [34], the gray matter of the cerebral cortex shows a distinctive developmental trajectory during adolescence [35,36,37], and myelination of white matter continues well into adulthood [38,39,40]. Many properties of the structural brain are highly heritable throughout the lifespan [41,42,43,44]. This includes their longitudinal change rates during adolescent brain development, which shares a common genetic etiology with psychiatric disorders and cognition [45,46,47,48,49,50,51]. Externalizing behavior has been linked to alterations in the structure and functioning of the brain; for reviews see [52,53,54,55,56]. Overall, various forms of externalizing behavior and psychopathology have been associated with reduced gray matter volume in the structures of the frontal and temporal cortex and the limbic system [57,58,59,60,61] and with increased white matter volume and reduced white matter integrity of the brain [62,63,64,65,66,67]. This pattern of reduced gray matter volume and increased white matter volume is often associated with more mature brain features as a result of synaptic pruning and continued myelination of the brain [68,69,70]. Several studies have reported an association between (longitudinal changes in) externalizing behavior with (longitudinal changes in) regional gray matter [71,72,73,74,75,76,77] and white matter structures [67,78,79,80]. However, the extent and even the direction of the association between externalizing behavior and brain structures differ between studies, especially in youth [54,75], which could be explained by the continuous development of the brain during childhood and adolescence that, in particular, affects studies on cross-sectional data. This emphasizes the need for longitudinal studies to analyze the association between brain and behavior within the context of their developmental trajectories in an individual [81].

Here, we report on a longitudinal study of adolescent twins and an older sibling from 112 families who were assessed around the ages 10, 13, and 18 years. Externalizing behavior was assessed by the Child Behavior Checklist (CBCL) and Youth Self Report (YSR), and gray matter volumes and white matter integrity (FA and MD) of the brain were assessed by structural magnetic resonance imaging (MRI) and diffusion tensor imaging (DTI). The longitudinal design allowed for the control of developmental effects within the individual, and to assess to what extent brain structures and externalizing behavior are associated, including the association between their longitudinal change rates. The inclusion of twins and siblings allowed for an application of the classical twin design to decompose variation in externalizing behavior and brain measures into genetic and environmental components, and to assess the extent to which shared genetic and environmental factors determine the covariation of externalizing behavior and these brain measures.

## 2. Results

### 2.1. Phenotypic Development of Externalizing Behavior

Endorsement of externalizing behavior on the Child Behavior Checklist (CBCL) and Youth Self Report (YSR) was low overall (Table 1; Appendix A), as is to be expected for a cohort of typically developing children. The twins and siblings were more likely to endorse items on the self-reported YSR than their parents reported on the CBCL (Table 1; Appendix A). Parents reported a slight decrease in externalizing behavior with age that is in contrast with the slight increase in externalizing behavior reported by their offspring (Figure 1). Differences in externalizing behavior between the sexes were negligible and were only significant for the externalizing scale on the YSR but not on the CBCL (Figure 1). Between assessments of the same instrument at the different ages, the phenotypic correlation is moderate to strong, but also indicates considerable individual variation between ages (correlation range CBCL: +0.52 to +0.64; correlation YSR: +0.46; Appendix A).

### 2.2. Heritability of Externalizing Behavior

The genetic influences on externalizing behavior were generally strong at all ages (*h*^2^ range: 42% to 88%; *p* < 0.030; Table 1) and the heritability estimates were similar across the ages for the parent-reported CBCL (*p* > 0.128). The heritability estimates for the externalizing score on the self-reported YSR dropped significantly at 18 years of age (*p* < 0.024). Influences from the common environment on externalizing scores were small across all ages and did not reach the significance threshold (*c*^2^ range: 0% to 9%; *p* > 0.237 [n.s.]; Table 1). The high heritability estimates were supported by the correlations within monozygotic (MZ), dizygotic (DZ) twin pairs, and twin-sibling pairs (Appendix A). A similar pattern of genetic and environmental influences existed for the untransformed scores, but with slightly lower heritability estimates and slightly higher common environmental influences (Appendix A).

Strong genetic influences were also found for the longitudinal changes between the assessments of externalizing behavior for the parent-reported CBCL ($h_{\Delta }^{2}$ range: 78% to 86%; *p* < 4.12 × 10^−4^; Table 1). The genetic influences on the longitudinal changes of externalizing behavior measured with the YSR did not reach significance ($h_{\Delta }^{2}$: 26%; *p* = 0.060 [n.s.]; Table 1). The analysis of the genetic correlation between assessments of externalizing behavior with the parent-reported CBCL at different ages showed substantial but incomplete overlap between genetic factors, indicating the presence of shared and distinct genetic factors influencing externalizing behavior at specific ages throughout adolescence (*r*_a_ range: +0.53 to +0.63; *p* < 3.73 × 10^−5^; incomplete pleiotropy *p* < 8.74 × 10^−4^; Appendix A). For the self-reported YSR, the genetic correlation suggested a single genetic factor influencing behavior at both ages (*r*_a_: +0.86; *p* = 7.12 × 10^−4^; incomplete pleiotropy *p* = 0.484 [n.s.]; Appendix A).

### 2.3. Phenotypic Development and Heritability of Brain Structures

We have previously described the developmental patterns and reported the heritability of brain structure and function throughout adolescence for this cohort; this included the longitudinal changes in brain structures that were also partially influenced by genetic factors [37,39,45,46,82,83,84,85,86,87].

The developmental pattern and heritability estimates of the six global brain measures are in line with our previous reports (Figure 2; Appendix A). Total brain volume, subcortical gray matter volume, cerebral white matter volume, and global mean fractional anisotropy and mean diffusivity continued to increase and/or level out nearing the end of adolescence, and cortical gray matter volume decreased steadily with age (*p* < 0.005). There were significant differences between the sexes, with higher gray matter volumes and global mean diffusivity but lower global mean fractional anisotropy for boys than for girls (*p* < 0.025). The heritability of global brain measures was high (*h*^2^ range: 34% to 93%; *p* < 0.001), with negligible effects from the common environment except for global mean diffusivity (*c*^2^ range: 10% to 28%; *p* < 0.025). Longitudinal changes in total brain volume were also heritable ($h_{\Delta }^{2}$ range: 33% to 64%; *p* < 0.013) and longitudinal changes in global mean diffusivity was influenced by the common environment ($c_{\Delta }^{2}$ range: 35% to 55%; *p* < 0.020).

The longitudinal profiles of the regional brain structures followed mostly similar patterns as the global brain structures (Appendix A). Most cortical regions showed a decrease in gray matter volume with age, with an occasional sex effect. In contrast, the gray matter volume of the amygdala and hippocampus slightly increased with age. The maturation of the mean fractional anisotropy of the white matter tracts started to slow down nearing the end of adolescence, whereas maturation of the mean diffusivity of the white matter tracts mostly increased throughout adolescence. Most regional brain measures were strongly influenced by genetics (*h*^2^ up to 84%; *p* < 0.05). On some occasions, influences from the common environment played a role as well (*c*^2^ up to 42%; *p* < 0.05). Genetics played a considerable role in the longitudinal change rates of gray matter volume and white matter integrity during adolescent development ($h_{\Delta }^{2}$ up to 67%; *p* < 0.05). Occasionally, there were also significant influences from the common environment on the longitudinal change rates ($c_{\Delta }^{2}$ up to 66%; *p* < 0.05). The associations between longitudinal assessments of the same brain measures indicated strongly overlapping genetic factors throughout adolescent brain development (Appendix A).

### 2.4. Assocations between Externalizing Behavior and Global Brain Structures

Significant associations between brain structures and externalizing behavior mainly occurred during mid-adolescence around 13 years of age. We note that the direction and magnitude of the associations were overall consistent throughout adolescence regardless of the significance of the associations and despite small variations between the instruments for measurement of externalizing behavior (Figure 3; Appendix A).

#### 2.4.1. Phenotypic Associations

Significant phenotypic associations between externalizing behavior and global brain structures were found primarily at 13 years of age (Figure 3; Appendix A). Global cortical gray matter volume was significantly associated with externalizing behavior on the CBCL (*r*_ph_ = −0.20; *p* = 0.012) and YSR (*r*_ph_ = −0.17; *p* = 0.038) at 13 years of age. Total cerebral white matter was significantly associated with externalizing behavior on the YSR at 13 years of age (*r*_ph_ = +0.20; *p* = 0.014). Global mean fractional anisotropy was significantly associated with externalizing behavior on the CBCL at 10 (*r*_ph_ = +0.18; *p* = 0.024) and 13 (*r*_ph_ = +0.20; *p* = 0.016) years of age and at 13 years of age for the YSR (*r*_ph_ = +0.18; *p* = 0.029). Global mean diffusivity was significantly associated with externalizing behavior on the CBCL at 13 years of age (*r*_ph_ = −0.20; *p* = 0.014), and at ages 13 (*r*_ph_ = −0.20; *p* = 0.016) and 18 (*r*_ph_ = −0.20; *p* = 0.010) years of age for the YSR. No significant associations were found between longitudinal change rates in externalizing behavior on the CBCL or YSR and change rates in global brain measures (Appendix A).

#### 2.4.2. Genetic Associations

Significant genetic associations between externalizing behavior and three global brain measures were found at 13 years of age (Figure 3; Appendix A). The association between global mean fractional anisotropy and externalizing behavior on the YSR at 13 years of age was partially due to the same additive genetic factor (*r*_a_ = +0.42; *p* = 0.038). Similarly, the association between global mean diffusivity and externalizing behavior on the CBCL was partially due to the same additive genetic factor (*r*_a_ = −0.27; *p* = 0.041) and YSR (*r*_a_ = −0.33; *p* = 0.031) at 13 years of age.

#### 2.4.3. Environmental Associations

The phenotypic associations between total cortical gray matter volume and externalizing behavior on the CBCL at 13 years of age were partially due to the same common environmental factor (*r*_c_ = −0.97; *p* = 0.026; Appendix A). In addition, the phenotypic association between the longitudinal change rates in global mean fractional anisotropy and changes in externalizing behavior on the CBCL between 10 and 18 years old were partially explained by the same unique environmental factor (*r*_e_ = +0.44; *p* = 0.032; Appendix A).

### 2.5. Assocations between Externalizing Behavior and Regional Brain Measures

A more nuanced pattern was suggested at the level of regional brain measures in an exploratory analysis (Appendix A), but results for regional brain structures did not survive correction for multiple comparisons and should be interpreted with caution.

## 3. Discussion

In this study, we measured externalizing behavior and brain measures in a longitudinal cohort at 10, 13, and 18 years old and tested to what extent the gray and white matter structural brain measures were associated with externalizing behavior. Given that the data were collected in twin families, we could also test the etiology of associations, i.e., the same genetic and environmental factors, influences both brain development and behavior. We found that a higher mean white matter fractional anisotropy (FA) and a lower mean white matter diffusivity (MD) were associated with more pronounced externalizing behavior during adolescence (*r*_ph_ up to −0.20/+0.20). These associations were due to the same additive genetic factors (*r*_a_ up to −0.33/+0.42) influencing both traits. Total brain volume was not significantly associated with externalizing behavior. Lower total gray matter volume and increased cerebral white matter volume were phenotypically, but not genetically, associated with more pronounced externalizing behavior around mid-adolescence (*r*_ph_ up to −0.20). Both externalizing behavior and white matter structure were partly heritable, and genetically correlated, which implies that during adolescent brain development genetic factors are involved in white matter integrity and are also involved in externalizing behavior.

The heritability of externalizing problems was influenced by genetics (*h*^2^ up to 88%); although a remarkable drop in heritability was observed for self-reported externalizing behavior around 18 years old. In parent-reported externalizing behavior during adolescence, a decrease in externalizing problems was observed, which was not seen in self-reports—though we note that these were collected twice, rather than three times. The decrease in parent-reported externalizing problems was influenced by genetics ($h_{\Delta }^{2}$ up to 86%). This discrepancy between parent- and self-reported behavior might reflect differences in reliability or construct validity but could also reflect the unique views of each rater on the behavior of the individual [88]. The differences between parent- and self-reported behavior were also observed in association with brain structures. Overall, we found that the phenotypic association between white matter integrity and externalizing behavior had the same direction and magnitude during adolescence but was most prominently present around 13 years old. However, the association disappeared around 18 years old for parent-reported externalizing behavior but remained present for self-reported externalizing behavior. Previous research in twins showed that longitudinal stability of externalizing behavior is partially due to a general age-independent genetic factor [26,88,89,90]. For brain structure and function, we have previously shown in the BrainSCALE cohort that an age-independent genetic factor also explains stability [37,45,85]. Thus, despite the developmental changes in both brain and behavior, there might be a more prominent overlap between the age-independent genetic factors for externalizing behavior and white matter integrity throughout adolescence.

We found that whole-brain global white matter integrity was both phenotypically and genotypically associated with externalizing behavior. White matter integrity has often been associated with behavior and cognition, with a more pronounced white matter integrity being associated with more mature behavior or higher cognitive skills [40,70,87,91]. The association of higher white matter integrity with more pronounced externalizing behavior has been reported before in children with conduct disorder and adults with psychopathic traits [63,66,92,93,94,95,96,97]. In general, in studies of cohorts with individuals who exhibit more extreme externalizing behaviors (e.g., conduct disorder, psychopathy, and incarcerated criminals and offenders), they tend to report decreased white matter integrity for those individuals with more externalizing behavior compared to the controls [52,98,99,100,101,102,103]. However, mixed results are reported for a broader range of externalizing behaviors, especially for studies in developmental cohorts [54,64,67,78,104], and it has been hypothesized that altered white matter integrity during development might depend on the presence or absence of callous (unemotional) traits co-occurring with the externalizing behavior [65,105,106,107,108]. In addition, we found that a larger global cortical gray matter volume was associated with less pronounced externalizing behavior around 13 years old, which was due to a correlated common environment factor. Cortical gray matter volume has been reported before in relation to externalizing behavior [57,58,59], including in a very large cohort of early adolescents around 9 to 10 years old [60,61], and in adults with callous or psychopathic traits [52,53]. Of note, we found that total brain volume and subcortical brain volume did not significantly associate, either phenotypically or genotypically, with externalizing behavior during adolescence. Thus, overall brain size could not explain the phenotypic and genotypic associations we found between externalizing behavior with global white matter integrity and cortical gray matter during adolescent development.

In our exploratory analysis of regional brain structures, a phenotypic association was most reliably and consistently found for the anterior thalamic radiation, the cingulum, forceps minor, and the bilateral corticospinal tract white matter integrity. We add that these associations represent correlated genetic influences. Similarly, for local gray matter volumes, the rostral middle frontal gyrus in particular was most reliably and consistently associated with externalizing behavior both phenotypically and genetically. However, these findings did not survive multiple comparison corrections and thus have to be interpreted with caution. This absence of significant findings at the regional brain level might be due to the correction for the effect of the global brain in our analyses, where the sum of small effects in regional brain structures contribute to an overall effect at the global brain level. Indeed, the associations between externalizing behavior and brain structure gray and white matter seem mostly related to individual differences for global brain structures. Our results suggest that the same genes are involved in both externalizing behavior and (global) brain structures, and that these brain structures could play a mediating role between the genetic liability and demonstration of externalizing behavior.

The quest to identify genetic variants related to externalizing behavior [109,110,111,112], brain structures [113,114,115], and longitudinal changes in brain structures [116] is a highly relevant and ongoing effort that requires much larger sample sizes than the current cohort provides [117]. However, the findings based on this extended twin cohort do reveal that such an endeavor could be relevant to pursue. Including neuroimaging-related phenotypes may aid in identifying subsets of genes associated with externalizing behavior that elucidate the biological pathways that are most relevant to the developmental trajectories of externalizing behavior, including those of relevance during childhood and adolescence.

There are some limitations in this study, including sample size. For a longitudinal neuroimaging study with three measurements, the sample size is decent, but it is only modest for genetic analyses of twins and siblings. A larger study sample may be required to detect the generally weak associations between brain structure and externalizing behavior [54,60,75], especially for the regional analysis of many local brain measures that require proper correction for the multiple comparisons to suppress the false positive rate. Second, this study was performed with an MR scanner at 1.5 Tesla. Higher MR field strengths may yield better tissue contrasts and/or allow for more advanced acquisition protocols. However, the field strength was intentionally not upgraded to higher field strengths to minimize the effects of scanner differences in longitudinal data acquisition. Third, we recognize that the externalizing problem assessment is based on self- and parent report and that agreement between them tends not to be high. It has been argued that this reflects each informant’s unique view on problem behavior [118]. Combining agreement and differences in ratings of different informants might constitute a possibly valuable source of information. Fourth, this study was performed in a longitudinal cohort of typically developing children and adolescents, showing a low overall endorsement of externalizing behavior, as is expected for the Dutch population [8,9]. Studies with the enrichment of clinical samples and extremes of the population might be more sensitive to detect changes in brain structures related to antisocial behavior.

## 4. Materials and Methods

### 4.1. Participants

This project is part of the longitudinal BrainSCALE study on the development of brain and cognition in twins and siblings [119], which is a collaborative project between the Netherlands Twin Register (NTR) [120,121,122] at the Vrije Universiteit (VU) Amsterdam and University Medical Center Utrecht (UMCU). The BrainSCALE cohort is a representative sample of typically-developing children from the Dutch population. A total of 112 families with twins and an older sibling participated in the study. The twins and siblings were assessed with a battery of cognitive and behavioral tests and extensive neuroimaging protocols when the twins were 9 years old [82]. Two additional waves of follow-up assessments were conducted with largely similar study protocols when the twins were 12 and 17 years old [37,46,85,86,87].

The BrainSCALE study was approved by the Central Committee on Research Involving Human Subjects of The Netherlands (CCMO), and the studies were performed in accordance with the Declaration of Helsinki. The children and their parents signed informed consent forms. Parents were financially compensated for travel expenses and children received a present or gift voucher at the end of the testing days. In addition, a summary of the cognition scores and a printed image of their T_1_-weighted brain MRI scan, when available, were provided afterwards.

Here, we analyzed the longitudinal data of 221 twins and 99 older siblings that had diffusion MRI scans and/or reports on externalizing behavior from the Child Behavior Checklist (CBCL) or Youth Self Report (YSR) available at any of the three assessments (Table 2). Both the twins and sibling groups were well balanced to investigate potential sex effects. The siblings were, on average, 2.7 years older than the twins. Structural T_1_-weighted and diffusion-weighted MRI scans and behavioral assessment by CBCL or YSR were available for most participants (Table 2). Fewer MRI scans were available at the second assessment due to the presence of dental braces that were incompatible with the magnetic field of the MR scanner. The Youth Self Report (YSR) was not collected at baseline assessment because the twins were below the age for which the YSR was developed (i.e., 11–18 years) [2].

No YSR was collected at baseline assessment #1. At assessment #1 and assessment #2, the CBCL was filled out by the mother. At assessment #3, the CBCL was filled out by either the mother (179; 74%), the father (55; 23%), or both parents (5; 2%), or unknown (2; <1%). The number of participants for the scores on the scales of the CBCL and YSR might be slightly lower due to missing items (average: 3%; range: 0% to 6%). Abbreviations (in alphabetical order): CBCL = Child Behavior Check List; F = female; M = male; MRI = magnetic resonance imaging; s.d.= standard deviation; YSR = Youth Self Report.

### 4.2. Externalizing Behavior

Externalizing behavior of the participants was reported by the parents on the Child Behavior Checklist (CBCL) and the participants on the Youth Self Report (YSR) [2]. Both instruments assess competencies and psychopathology (behavioral and emotional problems) in children and adolescents based on empirical evidence with high validity, including assessment for clinical disorders on the DSM-oriented scales [123]. The CBCL was distributed to the parents at all three assessments; the questionnaire was completed by the mother at the baseline and second assessment, and by either the mother (74%), the father (23%), or both parents simultaneously (2%) at the third assessment. The YSR was distributed to the twins and siblings at the second and third assessment because the twins were below the age for which the YSR was developed (i.e., 11–18 years) at the baseline assessment. Both instruments have largely similar items for the ‘Externalizing’ scale (Appendix A). However, three items on the CBCL were not available on the YSR (see Appendix A). In addition, three items were not available on the older 1999 version of the CBCL that was administered at the baseline assessment to rate the behavior of the older siblings and were excluded from analysis for the CBCL at all three assessments of the CBCL (see Appendix A). The items were scored on a scale of 0–2 (0: never; 1: occasionally; 2: often). The summary scores for the ‘Externalizing’ scale were computed by summing the scores of the relevant items (Appendix A). The scores of individual items with either incomplete, ineligible, or ambiguous responses were excluded from analysis, resulting in missing summary scores for some participants (average: 3%; range: 0% to 6%). Given that the families represent a typical cross-section of the Dutch population, the endorsement of externalizing behavior is low overall, and the prevalence of clinical disruptive behavioral disorders (DBD) in the participants (i.e., exceeding a raw sum score of approximately 18, depending on sex and age of the participant) is about 5 to 10%, which conforms with expectation within the Dutch population [8,9]. The distributions of the sum scores on the CBCL and YSR scales are positively skewed and may have inflated zero responses (mean skewness: +1.48; range skewness: +0.96 to +2.20; zero responses: 3% to 36%; Appendix A). A log-normal distribution described the data better than a normal distribution for the sum scores, which therefore were transformed with a 10-base logarithmic function (Appendix A).

### 4.3. Brain Imaging

#### 4.3.1. MRI Protocol

Longitudinal MRI scans of the brain were acquired at the University Medical Center Utrecht on 1.5 Tesla Philips Achieva scanners (Philips, Best, The Netherlands) using the same protocol to minimize the impact of unwanted variation between assessments. A three-dimensional anatomical T_1_-weighted scan (Spoiled Gradient Echo; TE = 4.6 ms; TR = 30 ms; flip angle 30°; 160–180 contiguous coronal slices of 1.2 mm; in-plane resolution 1 × 1 mm^2^; acquisition matrix 256 × 256) of the whole head was made of each individual. Two series of diffusion-weighted (DWI) scans with opposite phase encoding direction (Single Shot Echo Planar Imaging [SS-EPI]; 32 diffusion-weighted volumes with diffusion weighting b = 1000 s/mm^2^ and 32 noncollinear diffusion gradient directions; 4 diffusion-unweighted [b = 0 s/mm^2^] scans; echo time (TE) = 88 ms; repetition time (TR) = 9822 ms; parallel imaging sensitivity encoding (SENSE) factor 2.5; flip angle 90°; 60 transverse slices of 2.5 mm, no gap, field of view (FOV) 240 mm; 128 × 128 reconstruction matrix; 96 × 96 acquisition matrix, no cardiac gating) were acquired for an optimal signal-to-noise ratio.

#### 4.3.2. Volumetric Brain Measures

Volumetric brain measures were obtained with FreeSurfer version 5.3 [124]. The structural T_1_-weighted scans were processed with the cross-sectional pipeline using the default parameters, but with a custom brain mask for the individual scans. In this study, we limited ourselves to the subcortical structures and cortical regions of the frontal, temporal, and insular cortex based on previous research [55,56,60,125,126]; see Appendix A for details. In addition, the following global brain measures were included in the analysis: total brain volume, total cortical gray matter volume, total subcortical gray matter volume, and total cerebral white matter volume.

#### 4.3.3. White Matter Microstructural Integrity Brain Measures

The diffusion-weighted (DWI) scans were processed with the FMRIB Software Library (FSL) version 6.0.3 [127]. First, the two DWI scans were combined and corrected for possible gradient-induced distortions and motion-related displacement with *eddycorrect* [128]. The diffusion gradient vectors were rotated accordingly [129]. Next, a diffusion tensor model was fitted to the diffusion pattern at each voxel with *dtifit*, providing three eigenvectors (representing the three principal directions of diffusion) and the corresponding eigenvalues. Fractional anisotropy (FA) and mean diffusivity (MD) values were calculated at each voxel as a measure of microstructural directionality from the eigenvalues [130]. The FA and MD maps were aligned to the FA template of the John Hopkins University (JHU) white matter atlas with *fsl_reg* and *applywarp* using a non-linear transformation. Finally, mean FA and mean MD were extracted for all 20 major white matter tracts of the John Hopkins University (JHU) white matter tracts atlas provided with FSL; see Appendix A for details. In addition, a global measure of mean FA and mean MD for white matter in the brain was extracted for the combined tracts in the atlas.

### 4.4. Statistical Analysis

#### 4.4.1. Main Analysis

First, we analyzed the longitudinal profiles of the broad externalizing score on the Child Behavior Checklist (CBCL) and the Youth Self Report (YSR) by longitudinal genetic modeling of the twins and siblings (Figure 4; described in the next section) to determine the dynamics of the genetic and environmental factors influencing externalizing behavior throughout adolescence. We chose to analyze the scores on the CBCL and YSR separately because of the slight differences in items included on the instruments and the number of times each instrument was collected, and because each rater can have a unique view on problem behavior [118]. In all analyses, the log_10_-transformed scores of the externalizing behavior were corrected for sex and linear age effects as fixed effects in the longitudinal genetic model.

Next, we briefly summarize the longitudinal patterns in the global and regional brain measures and how they are influenced by genetic factors and the environment. The development and heritability of brain measures in the BrainSCALE cohort have been described elsewhere [37,39,45,46,82,83,84,85,86,87]. In all analyses, brain measures were corrected for sex and linear and quadratic age effects as fixed effects in the longitudinal genetic model. In addition, total (sub)cortical gray matter volume, total cortical white matter volume, and regional gray matter volumes and white matter integrity brain measures were corrected for their global brain measure (i.e., either total brain volume, global mean fractional anisotropy, or global mean diffusivity). Correcting the regional brain measures for the effect of global brain measures allows for the separation of region-specific effects from the effect reported in the analysis of the global brain measures.

Finally, we analyzed the phenotypic, genetic, and environmental associations between the externalizing scores and brain structure characteristics at three ages in a longitudinal genetic model including variables for both brain and behavior (Figure 4). The associations of their longitudinal change rates, i.e., measured by the difference in the behavioral score or brain measure between two assessments, was obtained from this longitudinal genetic model. The model was fitted to each combination of the broad externalizing score on the 2 questionnaires and the 6 global and 88 lateralized regional brain measures (21 cortical structures, 3 subcortical structures, and 11 white matter structures; for full list see: Appendix A).

#### 4.4.2. Genetic Modeling

Genetic modeling of twin and sibling data provides information on the extent that the variation of a trait in the population is explained by genetic factors [131,132]. Monozygotic (MZ) twins share (nearly) 100% of their genetic material and dizygotic (DZ) twins and full siblings share, on average, 50% of their segregating genes. The inclusion of these relatives into an extended twin design, i.e., twins and additional family members, enables for the decomposition of the phenotypic variance (*V*_P_) of a trait into three variance components: additive genetic (*V*_A_), common environmental (*V*_C_), and unique environmental (*V*_E_) components of variance. Additive genetic influences represent additive effects of multiple alleles at different loci across the genome that act upon the phenotypic trait. Common environment represents influences that are shared between twins and siblings from the same family and causes them to be more alike than children who grow up in different families. Unique environmental influences are not shared between family members and may include measurement error [133]. If MZ twins resemble each other more than DZ twins and siblings for a trait, then the hypothesis that the trait is influenced by genetic factors is supported. If both MZ and DZ twins are more alike in resemblance than expected based on genetics alone, common environment likely plays a role. This rationale for the analysis of univariate traits can be generalized to also analyze the covariance among multiple traits or multiple time points.

#### 4.4.3. Structural Equation Modeling

The genetic and non-genetic variance components can be estimated by maximum likelihood analyses in a genetic structural equation model (SEM) by specifying a phenotype to be influenced by latent additive genetic factors, and common and unique environmental factors. These factors represent unobserved or latent variables with unit variance where path coefficients, symbolized by *a*, *c*, and *e*, go from a latent variable to an observed phenotype. The model is identified by putting constraints on the correlation ***ρ***_A_ between the latent variable A of family members; ***ρ***_A_ = 1.0 for MZ twins and ***ρ***_A_ = 0.5 for DZ twins and twin-sibling pairs. The correlation ***ρ***_C_ between latent variable C of family members is constrained to ***ρ***_C_ = 1.0 for all twins and siblings from the same family. The latent variable E is uncorrelated between individuals by definition. The sum of the squared path coefficients *a*^2^, *c*^2^, and *e*^2^, equals the variance explained by *A*, *C*, and *E*, where the phenotypic variance (*V*) of a trait is *V* = *A* + *C* + *E* = *a*^2^ + *c*^2^ + *e*^2^. The heritability (*h*^2^) of the trait is estimated as the proportion of phenotypic variance (*V*) that is due to additive genetic variance (*A*); i.e., *h*^2^ = *a*^2^/(*a*^2^ + *c*^2^ + *e*^2^). Structural equation models were fitted to the data in OpenMx version 2.19.1 (https://openmx.ssri.psu.edu/, accessed on 30 January 2022) [134], a package for structural equation modeling in R version 4.0.3 (https://www.r-project.org/, accessed on 30 January 2022) [135]. Model fitting was performed on the raw data with full-information maximum likelihood (FIML).

#### 4.4.4. Genetic and Environmental Influences on Longitudinal Externalizing Behavior and Brain Measures

A longitudinal genetic model, with multiple measurements of the same trait acquired at different ages within the same individuals, allows for estimating the dynamics of genetic and environmental influences on traits over age. This model provides estimates of heritability at each age and the genetic and environmental correlations that explain the sources of stable variance between ages. Genetic correlations (*r*_a_) represent the extent to which the same genes influence a trait at different ages: $r_{a}\left(x,y\right)=cov\;A_{x,y}/\sqrt{var\;A_{x}\cdot var\;A_{y}}$; where $cov\;A_{x,y}$ is the genetic covariance between the trait *x* and *y*, and $var\;A_{x}$ and $var\;A_{y}$ are the genetic variances of the traits *x* and *y,* e.g., externalizing behavior at 10 and 13 years old, or externalizing behavior at 10 years old and total brain volume at 10 years old. The genetic correlation can tell if the genetic factor is completely shared (i.e., pleiotropy) or only partially shared (i.e., incomplete pleiotropy), where at least one of the traits is influenced by a second genetic factor unique to that trait. The longitudinal genetic model can also provide estimates on the heritability of the longitudinal change rates (e.g., changes in behavior or brain measures between repeated assessments) [47]. The heritability of change rates can be an indication of fluctuating influences of the same genetic factor over time, or the presence of novel genetic influences (i.e., genetic innovation) unique to a specific age [37,45,85]. Similarly, the extent to which the same environment influences a trait at different ages can be determined by the correlation of common environmental (*r*_c_) and unique environmental (*r*_e_) factors.

#### 4.4.5. Statistical Testing

The significance of the parameters in the longitudinal genetic model was determined by a log-likelihood ratio test by comparing the likelihood of the model with additional constraints on the parameters to the likelihood of the less constrained model. For bounded variance components, the difference in −2 times the log likelihood (−2LL) between models with a single constraint follows a 50:50 mixture of *χ*^2^ distributions with zero and one degree of freedom, effectively allowing *p*-values to be cut in half [136]. For the analysis of the regional brain measures, a correction for multiple comparisons was performed using the false discovery rate (FDR) method [137].

## 5. Conclusions

In conclusion, in this study, we have shown the existence of a shared genetic factor between externalizing behavior and brain structures during adolescent development. This is a step towards understanding the genetic etiology of externalizing behavior and the potential mediating role of brain structures. Knowing these would allow research to target key actors in developing diagnostic instruments or interventional therapies to prevent antisocial behavior during childhood and adolescence from escalating and to avoid criminal behavior or incarceration in later life.

## Figures and Tables

**Figure 1 ijms-23-03176-f001:**
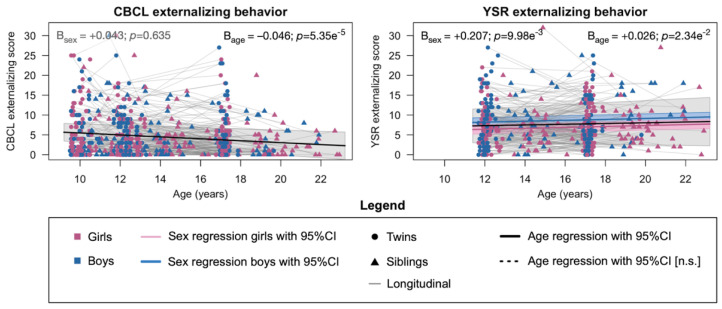
Phenotypic development of externalizing behavior throughout adolescence. Measurements are based on the externalizing scale of the Child Behavior Checklist and Youth Self Report. Data points are plotted for the untransformed scores on the y-axis. Estimates of sex and age regression coefficients and *p*-value were obtained from the longitudinal genetic model on the log_10_-transformed scores. Abbreviations (in alphabetical order): CBCL = child behavior checklist; CI = confidence interval; n.s. = not significant; YSR = youth self-report.

**Figure 2 ijms-23-03176-f002:**
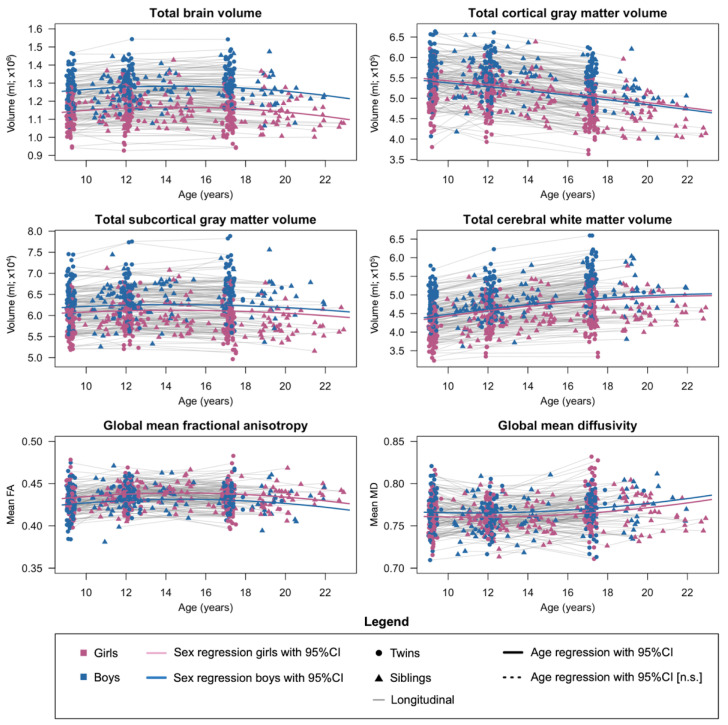
Development of global brain structures for boys and girls throughout adolescence. Sex effects for total (sub)cortical gray matter volume and cerebral white matter volume are plotted after simultaneous regression of total brain volume. The effect of sex was significant for all six global brain structures; the exact effect sizes and *p*-values of sex and age are reported in Appendix A. Abbreviation (in alphabetical order): CI = confidence interval; n.s. = not significant.

**Figure 3 ijms-23-03176-f003:**
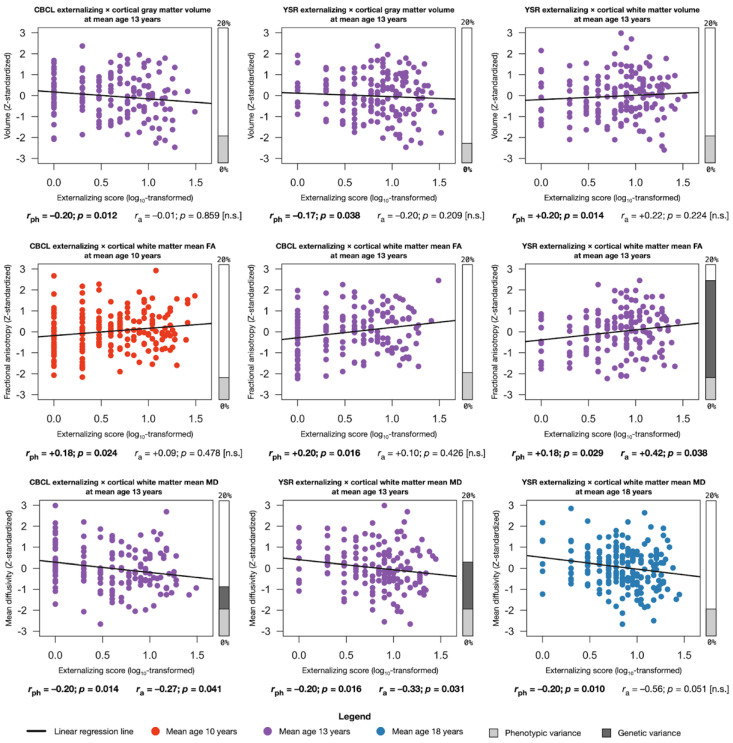
Associations between externalizing behavior and global brain structures throughout adolescence. Phenotypic association (*r*_ph_) of the log_10_-transformed scores of externalizing behavior on the CBCL or YSR questionnaire with Z-standardized measures of global brain structures (only listing uncorrected *p* < 0.05 significant associations). Bar charts display the total variance explained by the phenotypic (light gray) and genetic (dark gray) associations. Abbreviations (in alphabetical order): CBCL = Child Behavior Check List; FA = fractional anisotropy; MD = mean diffusivity; YSR = Youth Self Report.

**Figure 4 ijms-23-03176-f004:**
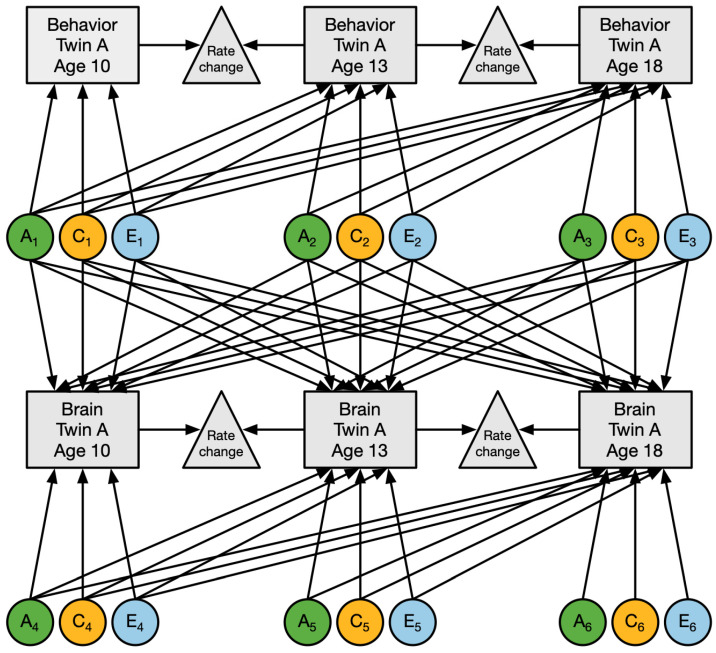
Longitudinal structural equation model applied to the longitudinal data on brain and behavior. Boxes represent observed measures of behavior and/or brain features; circles represent the latent factors for additive genetic (blue), common environment (orange), and unique environment (green). Path coefficients between the latent factors and observed measures are estimated by the model. For simplicity, this model is shown for only one member of the family. The full model includes three family members, where latent genetic factors between family members are correlated depending on their relationship (i.e., correlation of 1.0 for MZ twins, 0.5 for DZ twins and between twin and sibling pairs), common environmental factors between family members are always shared (i.e., correlation of 1.0), and unique environment factors are uncorrelated by definition.

**Table 1 ijms-23-03176-t001:** Externalizing behavior on the Child Behavior Checklist (CBCL) and Youth Self Report (YSR).

**Instrument**	**Age 10 Years**	**Age 13 Years**	**Age 18 Years**
CBCL	Score = 5.31 ± 6.1 [0; 30] *h*^2^ = **80% [57%; 92%]** *c*^2^ = 9% [0%; 29%] *e*^2^ = 12% [7%; 21%]	Score = 4.35 ± 5.1 [0; 30] *h*^2^ = **88% [70%; 96%]** *c*^2^ = 7% [0%; 25%] *e*^2^ = 5% [3%; 8%]	Score = 3.46 ± 4.9 [0; 27] *h*^2^ = **82% [63%; 91%]** *c*^2^ = 5% [0%; 22%] *e*^2^ = 13% [9%; 22%]
YSR	N/A	Score = 7.57 ± 6.0 [0; 32] *h*^2^ = **76% [58%; 86%]** *c*^2^ = 0% [0%; 10%] *e*^2^ = 24% [14%; 40%]	Score = 7.64 ± 5.3 [0; 27] *h*^2^ = **42% [15%; 59%]** *c*^2^ = 0% [0%;18%] *e*^2^ = 58% [41% ;80%]
**Instrument**	**Age 10 to 13 years**	**Age 13 to 18 years**	**Age 10 to 18 years**
CBCL	$h_{\Delta }^{2}$ = **86% [70%; 92%]** $c_{\Delta }^{2}$ = 0% [0%; 7%] $e_{\Delta }^{2}$ = **14% [7%; 29%]**	$h_{\Delta }^{2}$ = **84% [69%; 90%]** $c_{\Delta }^{2}$ = 0% [0%; 13%] $e_{\Delta }^{2}$ = **16% [10%; 26%]**	$h_{\Delta }^{2}$ = **78% [55%; 88%]** $c_{\Delta }^{2}$ = 1% [0%; 12%] $e_{\Delta }^{2}$ = **22% [12%; 38%]**
YSR	N/A	$h_{\Delta }^{2}$ = 26% [0%; 47%] $c_{\Delta }^{2}$ = 0% [0%; 17%] $e_{\Delta }^{2}$ = **74% [53%; 94%]**	N/A

The mean, standard deviation, and range [minimum; maximum] are reported for the untransformed scores of the externalizing scales on the CBCL and YSR. Heritability (*h*^2^), common environment (*c*^2^), unique environment (*e*^2^), and their 95% confidence intervals are reported for the longitudinal genetic model on the log_10_-transformed scores of the externalizing scale of the CBCL and YSR and their longitudinal change in scores ($h_{\Delta }^{2}$, $c_{\Delta }^{2}$, and $e_{\Delta }^{2}$). Heritability and common environment estimates printed in boldface are significant (*p* < 0.05); exact *p*-values are reported in Appendix A. No YSR is collected at baseline assessment #1 (10 years). Results for the genetic analysis on the untransformed scores are reported in Appendix A. Abbreviations (in alphabetical order): CBCL = Child Behavior Check List; YSR = Youth Self Report.

**Table 2 ijms-23-03176-t002:** Demographics and background information for the BrainSCALE twin cohort.

Trait	Assessment #1: Age 10 Years	Assessment #1: Age 13 Years	Assessment #1: Age 18 Years
Families	112	102	93
Participants	311 214 twins and 97 siblings	283 203 twins and 80 siblings	253 182 twins and 71 siblings
Sex	149 M and 162 F	133 M and 150 F	109 M and 144 F
Age ^1^	10.0 ± 1.4 [9.0; 15.0]	13.0 ± 1.5 [11.7; 18.0]	18.0 ± 1.4 [16.8; 22.9]
MRI scans	283	178	232
CBCL data	247	264	241
YSR data	N/A	253	243

^1^ Age in years; mean ± s.d. and range.

## Data Availability

Requests for access to the data and code used in this study should be directed to the corresponding author.

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
