# Peer review of "Multivariate Genetic Structure of Externalizing Behavior and Structural Brain Development in a Longitudinal Adolescent Twin Sample"

_ijms, 2022, doi:10.3390/ijms23063176_

Round 1
Reviewer 1 Report
The manuscript titled "Genetics of externalizing behavior and structural brain development in adolescence" deals with a very interesting topic. Understanding the etiology and development of externalizing behavior during childhood and adolescence is important for developing methods for early diagnosis and treatment to prevent escalation of antisocial behavior.
The paper is well-written and the methodology is explained in detail.
However, the results need to be reported more clearly.
In particular:
- Lines 171-173: values reported are relative to ℎ2? and ?2? and not to h2 and c2. Please, resolve this inconsistency.
- Line 231: it is reported that the value of ra is -0.27 while in Supplementary Table S9 it is reported as -0.26. Please, resolve this inconsistency.
- Lines 736-737: it is reported that the range of ?2? is 10% to 28% while in Supplementary Table S8 it is reported as 35% to 55%. Please, resolve this inconsistency.
In addition, the authors may consider the following minor comments:
- Line 25: please, specify the meaning of the abbreviations FA and MD.
- Lines 416-417: please, specify the meaning of the abbreviations TE, TR, SENSE and FOV.
- Line 430: please, specify the meaning of the abbreviation FSL.
- Line 548: please, specify the meaning of the abbreviation FDR.
Reviewer 2 Report
The present manuscript by Jalmar Teeuw et al., entitled:
Genetics of externalizing behavior and structural brain development in adolescence
aims to decipher the brain characteristics of adolescents showing externalizing behavior.
To this end authors analyzed brain imaging (MRI for brain volumes and white matter integrity) of adolescents showing externalizing behavior assessed by self-reported and parent-reported polls (i.e. Youth Self Report and Child Behavior Checklist, respectively). A heritability component was added, since investigated cohorts consisted of twins (monozygotic or dizygotic) and their older sibling (112 families measured longitudinally at ages 10, 13, and 18 years of the twins). Authors also analyzed the influence of common environment and unique environment of individuals in externalizing behavior considering the information provided in the polls.
This is a complex study combining multifaceted statistical methods.
However, several weak premises and methodological limitations restrain the conclusions of the investigation.
Among others:
- According to authors externalizing behavior in the studied cohorts was overall low (Table 2; Supplementary Figure S1), since it is a cohort of typically developing children in the Netherlands.
Thus, maybe the authors would need to compare results with cohorts showing more extreme externalizing behavior. For instance, with adolescents from a juvenile detention center or others.
Comparisons with cohorts from different countries could also be an approach, for instance including countries with high levels of externalizing behaviors in adolescents.
- The assessment of externalizing behavior in this study has a strong component of subjectivity, since it is just based on a self-reported toll and a parent reported toll.
Externalizing behavior might have been additionally assessed by an external evaluator, the same for all subjects (e.g. psychologist) to add a component of objectivity; And/or supported by facts (e.g. adolescents with related lawsuits, from a juvenile detention center, etc.).
The subjectivity of the assessment of externalizing behavior is an important limitation in this study: it is quite curious than results from the polls shows differences among parent-reported and self-reported evaluations, and for instance, according to parents externalizing behavior is more pronounced at age 10 years, than at age 13 or 18 (Table 2).
Thus, results may vary depending on: if the correlation is done considering the parent’s poll or the self-reported poll.
(E.g. L267: The decrease in parent-reported externalizing problems was influenced by genetics).
(L271: Differences between parent- and self-reported behavior were also observed in association with brain structures).
- To discuss about genetics of externalizing behavior: authors might want to include sequencing analysis and investigate alleles that could be involved in externalizing behavior. Comparing twins vs. siblings is a limited approach to discuss about genetics. In this sense, several sentences in the text may create misunderstanding to the reader.
E.g. L256 Lower total gray matter volume and increased cerebral white matter volume were phenotypically, but not genetically, associated with more pronounced externalizing behavior around mid-adolescence (rph up to –0.20).
Should the editors decide to publish this manuscript, authors may want to consider:
- To include further analysis/comparisons to address the issues mentioned above.
- To rewrite sentences regarding “genetics” that may create misunderstanding.
Other minor suggestions follow:
- L129: To explain abbreviations MZ and DZ.
- Table 2: Maximum score is 30? (e.g. 5.31±6.1 [0; 30]). Please clarify in figure legend and results.
- Other analysis could include sexual hormones in externalizing behavior.
- L204, 209, 224, 233, 241: first number in the subtitle should be “2.”
Round 2
Reviewer 2 Report
Authors have added some clarifying comments regarding the limitations of their study.
It is now easier to the reader to differentiate reliable data that may be of their interest.
L352: Please add Fourth in the listing: “Fourth, this study was performed…”.
Author Response
Thank you again for your review! We appreciate your help in improving the manuscript. We have revised the manuscript according to your suggestion.